# *Spop* deficiency impairs adipogenesis and promotes thermogenic capacity in mice

Qinghe Li[1]☉*, Yuhong Liu[1]☉, Yuanyuan Wang[2]☉, Qi Zhang[1]☉, Na Zhang[1], Danli Song[1], Fei Wang[3], Qianmei Gao[1], Yuxin Chen[2], Gaomeng Zhang[1], Jie Wen[1], Guiping Zhao[1]*, Li Chen[4,5]*, Yu Gao[2]*

1 Key Laboratory of Animal (Poultry) Genetics Breeding and Reproduction, Ministry of Agriculture and Rural Affairs, State Key Laboratory of Animal Biotech Breeding, Institute of Animal Science, Chinese Academy of Agricultural Sciences, Beijing, People's Republic of China, 2 School of Biological Sciences, Anhui Province Key Laboratory of Translational Cancer Research, Bengbu Medical University, Bengbu, China, 3 Key Laboratory of Animal Genetics, Breeding and Reproduction of Shaanxi Province, College of Animal Science and Technology, Northwest A&F University, Yangling, China, 4 Institute of Animal Science & Veterinary, Zhejiang Academy of Agricultural Sciences, Hangzhou, China, 5 Xianghu Laboratory, Hangzhou, China

☉ These authors contributed equally to this work.
* liqinghe@caas.cn (QL); zhaoguiping@caas.cn (GZ); chenli@zaas.ac.cn (LC); gaoyu@bbmc.edu.cn (YG)

**Data Availability Statement:** The datasets included in this study can be found in online repositories. Listed below are repository websites and accession numbers: http://www.ncbi.nlm.nih.gov/bioproject/1191071, PRJNA1191071 for RNA-

## Abstract

As the adaptor protein that determines substrate specificity of the Cul3-SPOP-Rbx1 E3 ligase complex, SPOP is involved in numerous biological processes. However, its physiological connections with adipogenesis and thermogenesis remain poorly understood. In the current study, we report that the conditional knockout of *Spop* in mice results in substantial changes in protein expression, including the upregulation of a critical factor associated with thermogenesis, UCP1. Loss of SPOP also led to defects in body weight gain. In addition, conditional knockout mice exhibited resistance to high-fat-diet-induced obesity. Proteomics analysis found that proteins upregulated in the knockout mice are primarily enriched for functions in glycolysis/gluconeogenesis, oxidative phosphorylation, and thermogenesis. Furthermore, *Spop* knockout mice were more resilient during cold tolerance assay compared with the wild-type controls. Finally, the knockout of SPOP efficiently impaired adipogenesis in primary preadipocytes and the expression of associated genes. Collectively, these findings demonstrate the critical roles of SPOP in regulating adipogenesis and thermogenic capacity in mice.

## Author summary

Excessive energy intake and insufficient physical activity are often considered to be the main reasons for obesity. An imbalance between energy intake and energy expenditure has a critical impact on reducing obesity. SPOP is an E3 ligase adaptor with diverse functions, including tumorigenesis, innate immune response, cell apoptosis, X chromosome inactivation, cellular senescence, and animal development. By using a *Spop* deficient mouse model, we showed that *Spop* deficiency resulted in a defect in the increase of body weight, accompanied by alterations in the expression of genes associated with glycolysis,

seq in the BAT of mice; https://ngdc.cncb.ac.cn/omix/release/OMIX008027 for proteomics analysis data of the kidney and liver of mice.

**Funding:** This work was supported by Biological Breeding-National Science and Technology Major Project (2023ZD0405302 to QL), Innovation Program of Chinese Academy of Agricultural Sciences (CAAS-CSAB-202401 to QL), Anhui Excellent Young Teachers Training Program (YQYB2024039 to YW), the Key Project of Natural Science Foundation of Anhui Provincial Department of Education (2023AH052002 to YW), the NSFC Incubation Program of Bengbu Medical University (2023byfy009 to YG) and the Hangzhou Science and Technology Development Program (202203A09 to LC). The funders had no role in study design, data collection and analysis, decision to publish, or preparation of the manuscript.

**Competing interests:** The authors have declared that no competing interests exist.

gluconeogenesis and thermogenesis. Meanwhile, *Spop* deficiency led to increased cold tolerance and attenuated fat deposition. Taken together, these findings demonstrate the critical role of *Spop* in the manipulation of adipogenesis and thermogenic capacity in mice.

## Introduction

Obesity traditionally refers to an excess of body fat, which condition is currently increasing epidemically. Its associated metabolic morbidities constitute a major health burden of modern society and cause comorbidities such as diabetes, cardiovascular disease, and cancer [1]. Individuals seeking to combat obesity have two approaches to apply: decrease energy intake or increase energy expenditure. Food intake and preference, such as for a high-calorie diet, are regulated by central mechanisms, while energy dissipation takes place mainly in thermogenic adipose tissue [2].

Adipose tissue plays a critical role in regulating mammalian energy metabolism and body weight maintenance. There are three major types of adipocytes found in most mammals, white, beige, and brown. White adipose tissue (WAT) is crucial as a fat-storing organ, providing long-term storage of surplus energy in the form of triacylglycerol and releasing free fatty acids during energy deprivation [3]. Brown adipose tissue (BAT) promotes total body energy expenditure and attenuates adiposity through dissipating energy via non-shivering thermogenesis [4]. In addition to the classical BAT, brown-like multilocular beige/brite adipocytes within WAT can be activated by exposure to cold, pharmacological treatment, and other stimuli [5]. The thermogenic capacity of BAT relies on its high mitochondrial density and the activity of the uncoupling protein UCP1 [6]. A proton transporter uniquely expressed in BAT and beige fat, UCP1 uncouples the mitochondrial proton gradient from ATP synthesis and thereby dissipates energy as heat [7,8]. Notably, BAT activity varies significantly among individuals, but is especially low in older and obese adults [8]. Therefore, unraveling the regulatory mechanisms that underlie the adipogenesis and thermogenic gene program is essential for developing new approaches to combat obesity and related metabolic disorders.

Thermogenesis and adipogenesis are under strict regulation *in vivo*. The first adipocyte-specific transcription factor to be identified, peroxisome proliferator-activated receptor γ (PPARγ), functions as a key regulator of adipocyte differentiation and is modulated by external stimulus [9]. PPARγ binds PPARc coactivator 1a (Pgc-1α) to induce thermogenic gene expression and adaptive thermogenesis [10]. Pgc-1α interacts with transcription factors that control thermogenic gene expression in BAT, and the knockdown of *Pgc-1α* has demonstrated that its function is essential for brown fat thermogenesis but not differentiation [11].

Speckle-type POZ (pox virus and zinc finger protein) protein (SPOP) was first identified as a novel nuclear antigen in serum from a scleroderma patient [12]. As an adaptor for the Cul3-RBX1 E3 ubiquitin ligase complex, SPOP determines the substrate specificity of the complex via binding through its N-terminal meprin domain and traf homology (MATH) domain [13]. SPOP has been revealed to play critical roles in the regulation of many biological processes, including tumorigenesis, innate immune response, cell apoptosis, X chromosome inactivation, cellular senescence, and animal development [14–17]. Moreover, recent studies have indicated SPOP to be a frequently mutated hotspot in several cancer types such as primary prostate, kidney, and endometrial cancer [15,18]. Most of the mutations identified thus far affect evolutionarily conserved residues in the substate-binding MATH domain, indicating that the alteration of its substrate recognition could promote the development of cancer.

While our understanding has expanded regarding the diverse biological functions in which SPOP operates, little is yet known about how its involvement with thermogenesis and adipogenesis. In the present study, we uncover for the first time the critical role of SPOP in regulation of adipogenesis and thermogenesis using *Spop* knockout mice as a model.

## Results

### Mapping of differentially expressed proteins in *Spop* conditional knockout mice

We generated an *Spop* conditional knockout (CKO) mouse model using the Cre-LoxP recombination in our previous study [19]. Western blotting showed that SPOP was efficiently knocked down in these mice (Fig 1A and 1B). To investigate the potential role of SPOP in the regulation of translational output, we mapped genome-wide protein expression in the kidney of the *Spop* CKO mice using 4D label-free quantitative proteomics. A total of 168 upregulated and 106 downregulated proteins were identified relative to the wild-type control with the criteria of $P < 0.05$ and fold change $> 1.5$ (Fig 1C and S1 Table). Gene Ontology analysis revealed that these differentially expressed proteins were enriched for genes involved in metabolic processes, anion transport, catabolic processes, and biosynthetic processes (Fig 1D). KEGG pathway analysis likewise indicated them to be significantly enriched in metabolism pathways, including regulation of bile secretion, lipolysis in adipocytes, fatty acid biosynthesis, fatty acid metabolism, fatty acid degradation, glycerolipid metabolism, and metabolism activities of cytochrome P450 (Fig 1E). The protein UCP1 stood out as particularly noteworthy, being a critical factor involved in adaptive thermogenesis in interscapular brown adipose tissue and significantly upregulated in the *Spop* CKO mice (Fig 1F). Ucp1 is predominantly expressed in brown adipose tissue; however, microarray assays in the mouse kidney have shown that it was also expressed in the renal tubular epithelial cells of normal kidneys, an expression that was dramatically reduced after acute kidney injury [20].

### Knockout of *Spop* in mice leads to defects in body weight gain

To determine whether knocking out of SPOP affects body composition, we fed the *Spop* CKO mice with a chow diet and monitored their body weights for ten weeks. The wild-type (WT) mice gained more than 50% more body weight during this period than did the CKO mice ($P < 0.05$, Figs 2A and S1), and were distinctly larger in size (Fig 2B). We also measured organ weights and found that most of the weights of the peripheral organs remained unchanged except the weights of the spleen and BAT decreased in the CKO mice (Fig 2C). We also measured the body composition of the mice and found that the lean mass dropped slightly, while there was a stronger decrease of fat mass in the *Spop* CKO mice compared with the wild-type controls (Fig 2D).

Abatement of adipose lipid storage is usually associated with disorders of circulating glucose and triglyceride (TG). Assaying TG in ten-week-old mice revealed the *Spop* CKO mice to display significant lower levels of both compared to the WT controls (Fig 2E). Histological analysis revealed markedly more lipid droplets in the liver of the WT mice (Fig 2F), indicating weaker deposition of lipids in the *Spop* CKO mice. H&E staining revealed the BAT of the *Spop* CKO mice to feature an obvious decrease in the deposition of lipids compared with the WT controls (Fig 2F).

To examine the effects of *Spop* knockout on transcriptional output in the BAT, we first performed genome-wide gene-expression analysis by RNA sequencing (RNA-seq). The transcriptomic profiles of the CKO mice differed substantially from those of wild-type controls, with a

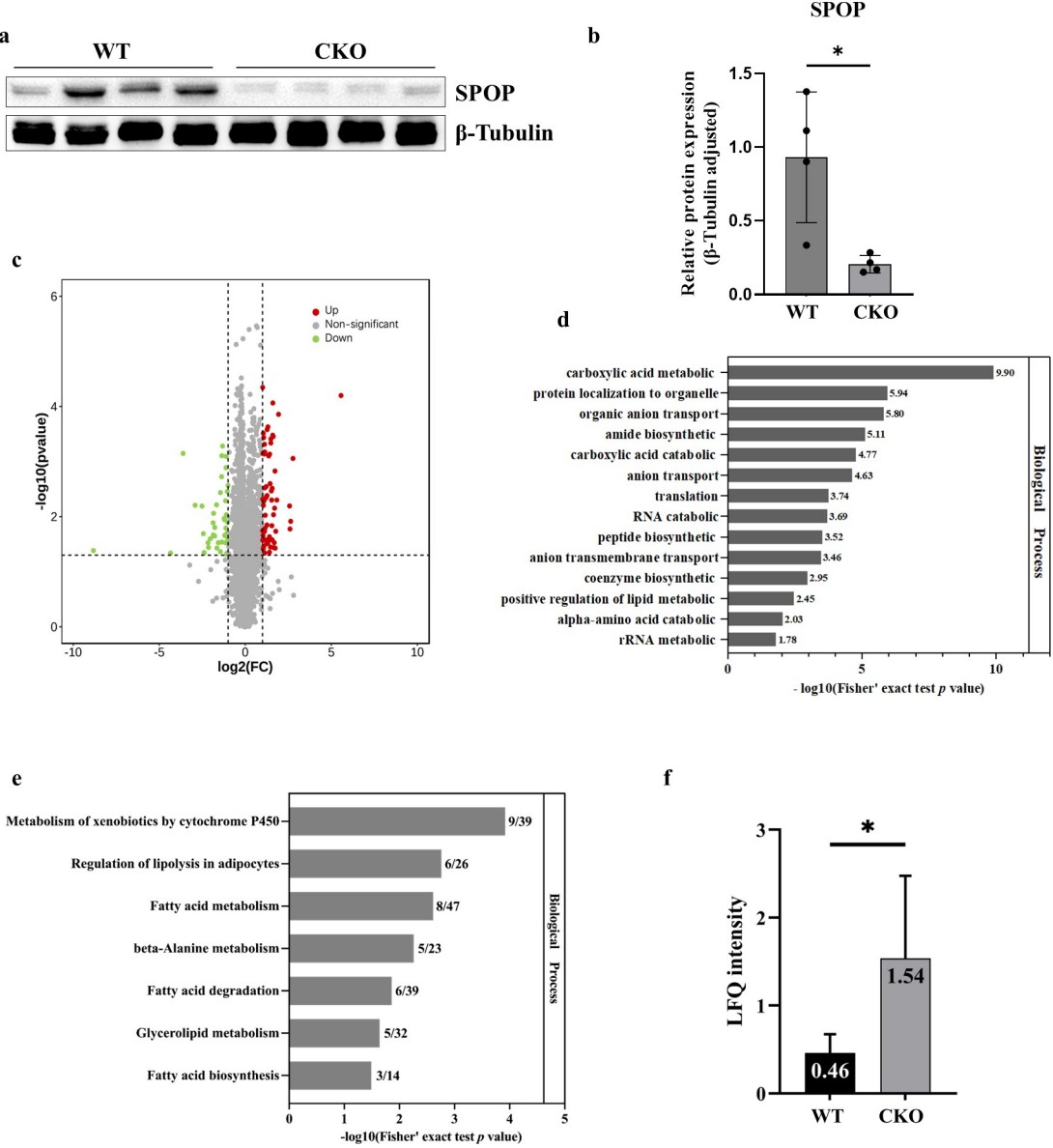

**Fig 1.** a, Immunoblot analysis of SPOP in the kidneys of *Spop* knockout and wild-type (WT) mice. b, Data quantification of panel (a). c, Volcano plots showing differentially expressed genes in the mouse kidney (CKO versus WT, n = 274). d, e, Gene Ontology (d) and KEGG pathway (e) Enrichment results for differentially expressed genes. f, The abundance of UCP1 in the kidney of CKO and WT mice as examined by Mass Spectrometry.

total of 77 genes downregulated and 44 genes upregulated (S2 Table). KEGG pathway analysis revealed the differentially expressed genes to be enriched for genes involved in metabolic pathways such as thermogenesis, non-alcoholic fatty acid liver disease, glucagon signaling, and adipocytokine signaling (S2 Fig). A few key upregulated genes were associated with thermogenesis (*Bmp8b* and *Fgf21*), others with the activity of the mitochondrial respiratory chain (*Cox6c2* and *Ndufb2*), and *Pik3r1* with lipolysis in adipocytes (Fig 2G). We also assessed the impact of SPOP deficiency on bioenergetics in BAT punches isolated from the *Spop* CKO and WT mice. As expected, citrate synthase activity was significantly increased in the BAT of *Spop* CKO mice (Fig 2H).

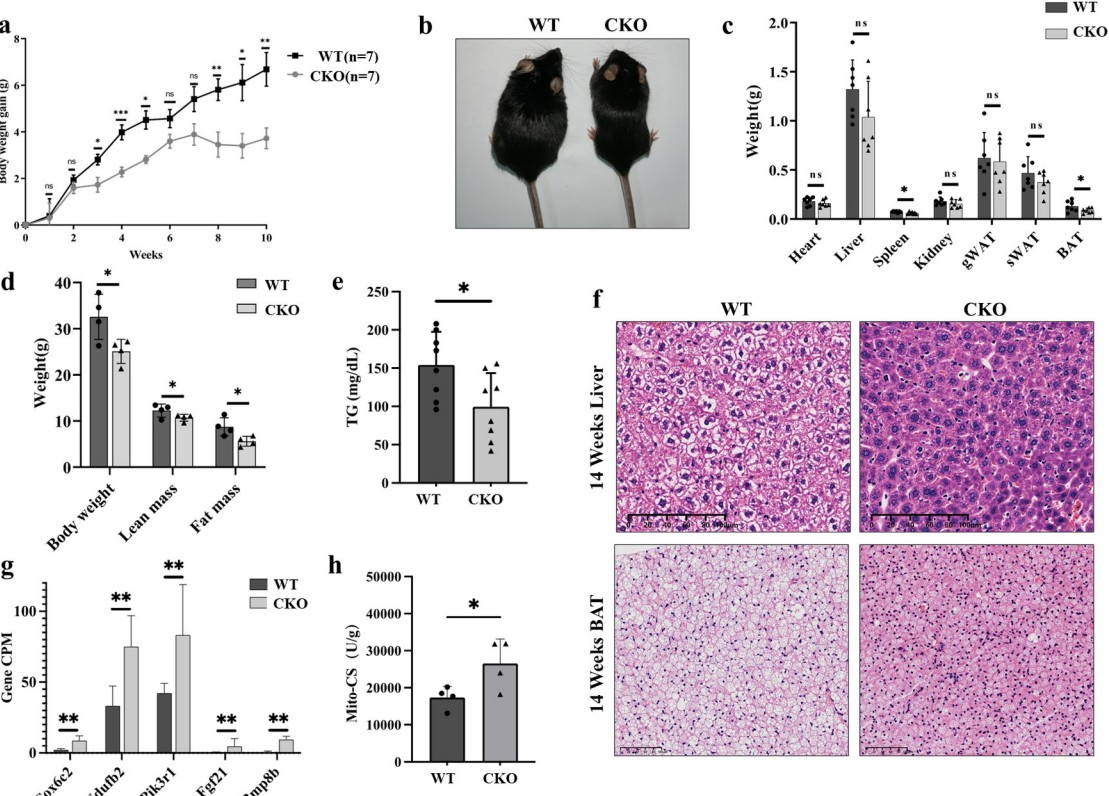

**Fig 2.** a, Body weight of the male WT and CKO mice fed a chow diet (n = 7). b, Representative photographs of the WT and CKO mice. c, Organ weight of the WT and CKO mice. d, body composition of the CKO and WT mice fed with a chow diet. e, Level of triacylglycerol in the serum of 14-week-old WT and CKO mice (n = 8). f, H&E staining of liver and BAT from 14-week-old WT and CKO mice. Scale bars as indicated in the figure. g, Expression of key genes associated with adipogenesis and thermogenesis in BAT from the CKO and WT mice. h, Mito-Citrate synthase activity (U/g) in the BAT of 14-week-old WT and CKO mice.

## *Spop* conditional knockout mice are resistant to diet-induced obesity

Next we investigated the effect of *Spop* knockout on mice fed a high-fat diet (HFD). Both the WT and CKO mice were fed an HFD for six continuous weeks. The resultant body weight gain in the *Spop* CKO mice was nearly 60% less than that observed in the WT mice despite the slight difference in feed intake (Figs 3A and S3). We also measured the weight of the organs and found that the weights of the liver, WAT and BAT decreased in the *Spop* CKO mice (Fig 3B). As for circulating glucose, triglyceride and cholesterol levels, we observed significantly lower serum glucose, TG and cholesterol levels in the CKO mice (Fig 3C). H&E staining further revealed fewer lipid droplets in the livers, BAT and gWAT of the *Spop* CKO mice (Fig 3D). Typically, weaker expression of PPARγ was observed in the BAT of *Spop* CKO mice (Fig 3E and 3F).

To explore the potential molecular mechanism of the *Spop* CKO-induced resistance to obesity, we profiled the protein expression in the liver by mass spectrometry. In total, we identified 330 differentially expressed proteins between the *Spop* CKO and WT mice, including 149 downregulated and 181 upregulated proteins (S3 Table). Consistent with the fat loss in the CKO mice, KEGG pathway analysis revealed the upregulated proteins to be primarily enriched in glycolysis/gluconeogenesis, oxidative phosphorylation, and thermogenesis (Fig 3G). Meanwhile, downregulated proteins were mostly enriched in fatty acid biosynthesis, glucagon signaling, and other metabolic pathways such as pentose phosphate and galactose metabolism

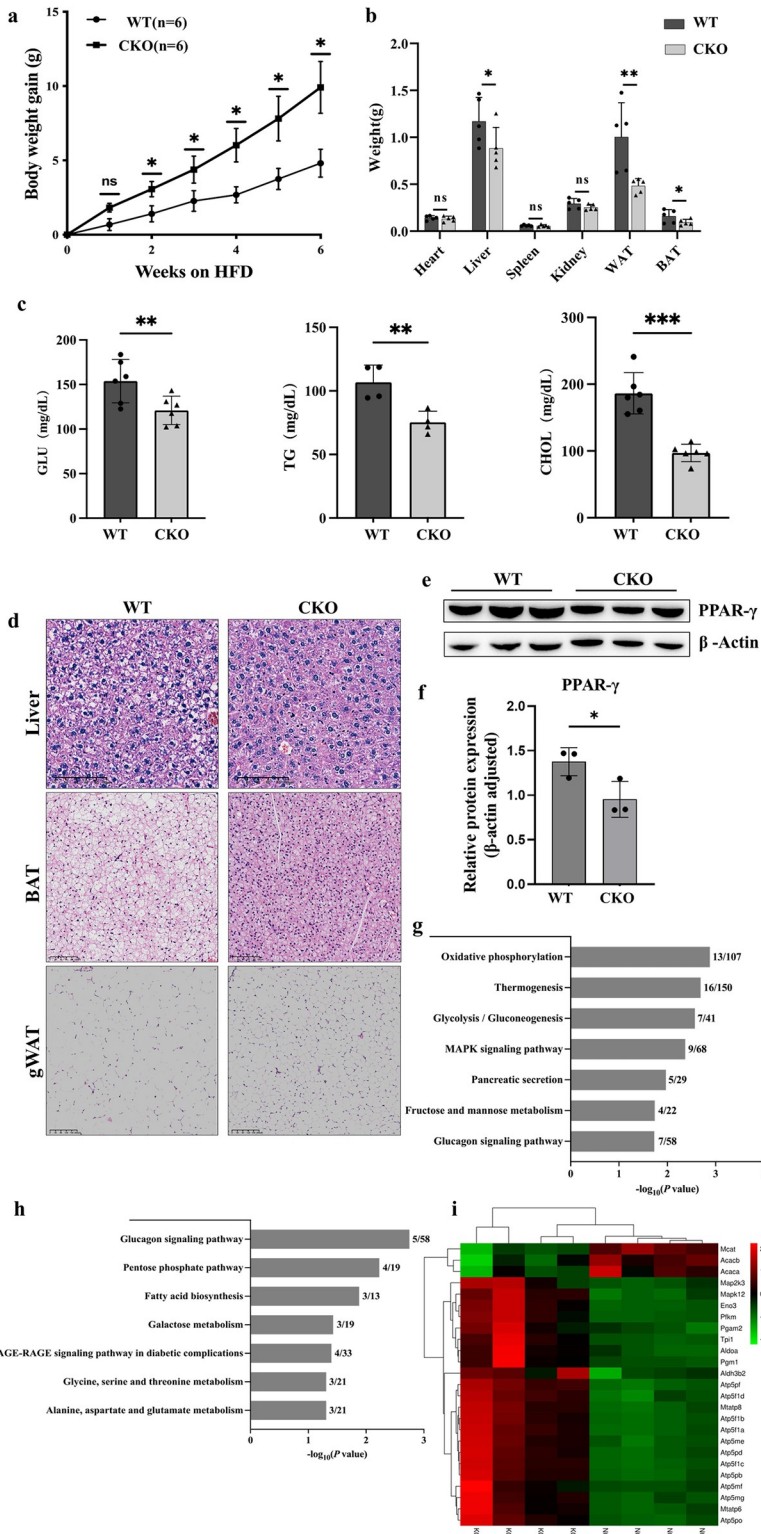

**Fig 3.** a, b, Body weight (a) and organ weight (b) of the male WT and CKO mice fed a high-fat diet (HFD) (n = 5). 0 week means the start of the HFD treatment, at eight weeks old. c, Levels of glucose, triacylglycerol and cholesterol in the serum of 14-week-old WT and CKO mice fed an HFD (n = 6). d, H&E staining of liver, BAT and gWAT from 14-week-old WT and CKO mice fed an HFD diet. Scale bars as indicated in the figure. e, the expression of PPARγ in the BAT of the CKO and WT mice as detected by western blot. f, Data quantification of panel (e). g, h, KEGG pathway

enrichment of up- (g) and down-regulated (h) proteins in the liver of the WT and CKO mice fed with an HFD. i, Heatmap of selected differentially expressed genes between the WT and CKO mice fed with an HFD.

(Fig 3H). Notably, we found several known proteins that had critical impacts on fatty acid synthesis, including malonyl-CoA-acyl carrier protein transacylase (MCAT) which catalyzed the transfer of a malonyl group from malonyl-CoA to the mitochondrial acyl carrier protein, acetyl-CoA carboxylase beta (ACACB), and acetyl-CoA carboxylase alpha (ACACA) which functioned as the rate-limiting step in fatty acid synthesis, to be significantly downregulated in the *Spop* CKO mice. We also found several ATPase components involved in oxidative phosphorylation to be upregulated in the *Spop* CKO mice (Fig 3I). We also quantified mRNA levels of two critical genes involved in fatty acid metabolism in the BAT, our results indicated that the expression of the transcription factor Pgc1α which was associated with lipid oxidation was upregulated in the *Spop* CKO mice, while the expression of fatty acid synthase (FASN) which was a vital enzyme in fatty acid synthesis was downregulated (S4 Fig).

## *SPOP* conditional knockout mice are more cold-tolerant

To examine the effect of SPOP deficiency on thermogenic capacity, we subjected the *Spop* CKO mice to acute cold exposure at 4°C. The wild-type mice were substantially less cold-tolerant than the CKO mice, as their body temperature started to drop significantly after 3 hours of cold challenge (Fig 4A). After 9 hours of cold exposure, the core body temperature of the CKO mice was more stable compared to the beginning, while the body temperature of the WT mice was 3°C lower than that of the CKO mice. Infrared thermal images also indicated that the *Spop* CKO mice exhibited higher surface temperatures than the wild-type mice (Fig 4B). We next examined the expression of UCP1 of the mice and found higher expression of UCP1 in the BAT of the *Spop* CKO mice (Fig 4C and 4D). These results provide *in vivo* evidence of the interference of SPOP for cold-induced thermogenesis.

To further explore the underlying mechanism explaining why *Spop* knock-out mice are more cold-tolerant, we used metabolic cages to profile the metabolic responses of the WT and *Spop* CKO mice using the Comprehensive Lab Animal Monitoring System (CLAMS). $O_2$ consumption ($VO_2$) and $CO_2$ ($VCO_2$) release were significantly enhanced in the CKO mice under normal conditions (Fig 4E and 4F), indicating that the *Spop* CKO mice had higher respiratory activity than the wild type. The respiratory exchange ratio (RER) of the CKO mice was also significantly lower compared with the WT mice (S5 Fig), implying that the CKO mice preferred to use fatty acids as the supply of energy or were incapable of switching from fatty acids to carbohydrates.

## Depletion of SPOP inhibits adipocyte differentiation

To investigate whether SPOP affects adipocyte differentiation, we isolated primary preadipocytes from the *Spop* CKO and wild-type mice. Western blot results showed that the expression of SPOP was effectively inhibited in the preadipocytes of *Spop* CKO mice (Fig 5A and 5B). Oil red O staining results showed that the formation of lipid droplets was dramatically impaired after SPOP knockout (Fig 5C). As a result, the triglyceride level dropped in SPOP depleted adipocytes (Fig 5D). Furthermore, differentiated cells were collected to examine the effect of knockout of *Spop* on adipogenesis, the expression of CEBPβ which was associated with adipogenesis and PCNA and CYCLIND1 which were associated with cell proliferation were significantly decreased (Fig 5E).

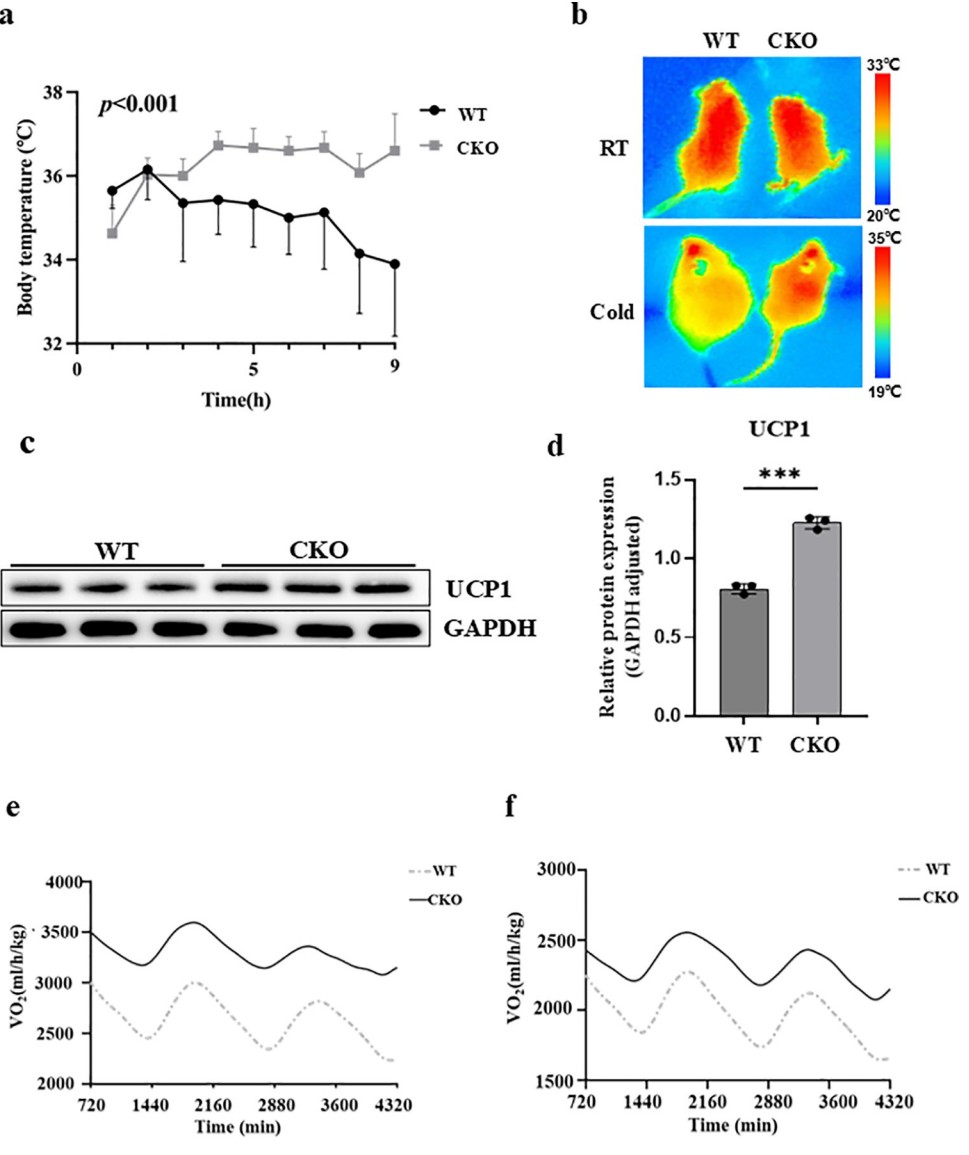

**Fig 4.** a, Body (rectum) temperature of the WT and CKO mice under acute cold exposure at 4°C. b, Representative infrared pictures of the 2-month-old mice collected after 3 hours cold exposure. c, the protein level of UCP1 in the BAT of the WT and CKO mice. d, Data quantification of panel (c). e, f, $VO_2$ (c) and $VCO_2$ (d) of mice at the indicated times under normal conditions.

## Discussion

SPOP is associated with many biological functions through targeting a variety of substrates. It is reasonable to speculate that more functions of SPOP will be uncovered as new substrates are discovered. Several groups, including ours, have revealed SPOP to have critical effects on the regulation of the innate immune response and hematopoiesis [19,21–23]. Using a mouse model of conditional *Spop* knockout, we surprisingly found that SPOP also plays crucial roles in promoting adipogenesis and impairing thermogenesis. Our findings expand the role of SPOP and uncover its association with adipogenesis and thermogenesis for the first time.

In the present study, we identify SPOP involvement in the regulation of thermogenesis to occur through manipulating associated genes including *Ucp1*, *Cox6c2*, *Ndufb2*, and *Fgf21*. We

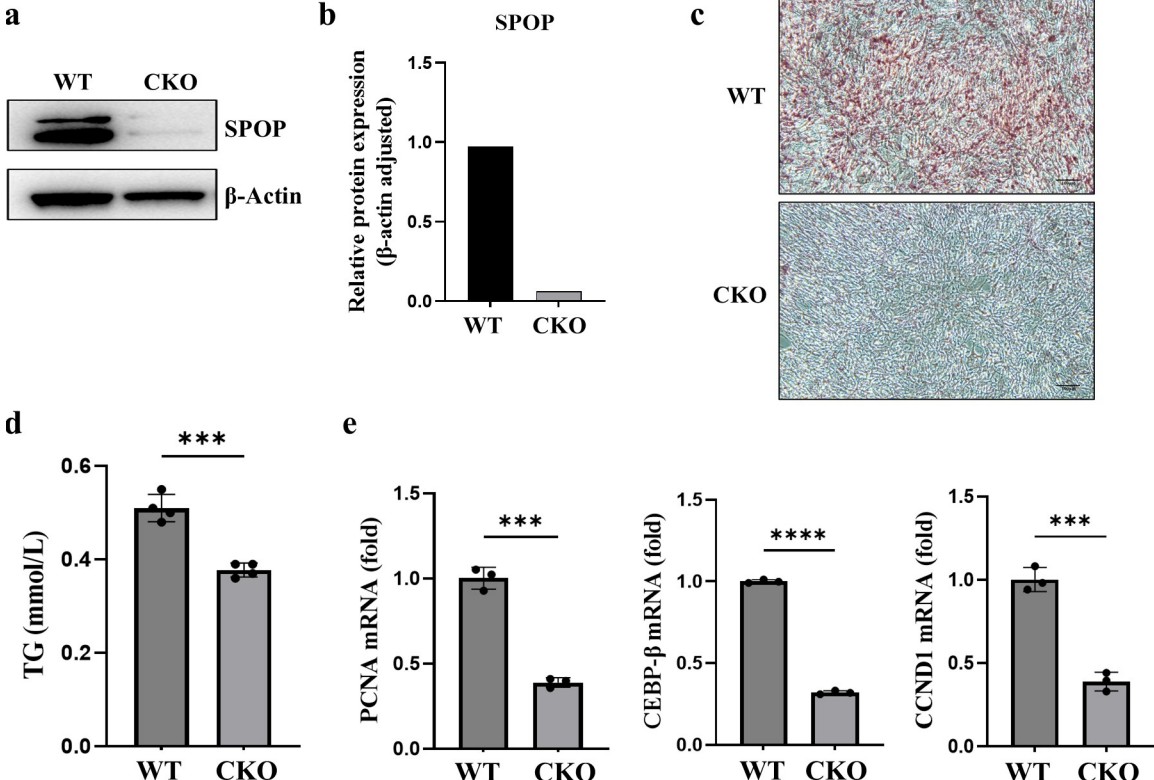

**Fig 5. SPOP is required for adipocyte differentiation.** a, the knockdown efficiency of SPOP as detected by western blotting. b, Data quantification of panel (a). c, Primary adipocytes were differentiated into mature adipocytes. Oil red O staining of the lipid droplet showed that the efficiency of adipocytes differentiation in the *SPOP* knockout cells was dramatically decreased compared with the wild-type control cells. d, The triglyceride levels in the primary adipocytes after *SPOP* knockout. e, The mRNA expression levels of genes involved in adipogenesis and proliferation.

firstly observed through a proteomics analysis that *Spop* knockout in mice leads to the dysfunction of genes associated with fatty acid metabolism and energy expenditure. In particular, UCP1, which is a major regulator of brown and beige adipocyte energy expenditure and metabolic homeostasis [24], was upregulated more than three-fold. UCP1 is also known to contribute to protection against obesity [25]. Based on this finding, we speculate that SPOP might function in energy expenditure and fatty acid metabolism. When fed with a standard chow diet, the *Spop* knockout mice had significantly lower body weights than their control littermates, accompanied by lighter lipid deposition in the liver.

Subsequently, gene expression analysis uncovered higher expression of BAT signature genes in the *Spop* CKO mice. Interestingly, the BAT of these CKO mice appears to be morphologically whitened even as it more highly expresses BAT signature genes. A previous study reported that the hypertrophy of brown adipocytes may simultaneously increase lipid storage and thermogenic capacities [26]. The greater lipid storage in the BAT of the *Spop* CKO mice might serve as an adaptive mechanism to provide sufficient lipid supply to match a greater rate of thermogenesis.

We observed a series of genes associated with cytochrome-c oxidase activity, heat production, glucose uptake, glycogen synthesis, and thermogenesis in brown adipose tissue were morehighly expressed in the BAT of the *Spop* CKO mice. For example, FGF21 is established as a stress-responsive cytokine that regulates energy balance by increasing energy expenditure [27,28]. Transgenic overexpression and pharmacological administration of FGF21 under

obesogenic conditions has been shown to protect animals from diet-induced obesity and improved systemic glucose homeostasis [29,30]. Two key genes of the mitochondrial respiratory chain, *Cox6c* and *Ndufb2*, likewise showed significantly higher expression in the *Spop* CKO mice; notably, mitochondrial respiratory chain activity is associated with heat production [31,32]. *Bmp8b*, a gene primarily expressed in the BAT and testes, directly regulates mouse thermogenesis and energy balance in partnership with hypothalamic AMPK [33,34]. *Bmp8b* knockout mice display impaired thermogenesis capacity and are susceptible to diet-induced obesity [33,34]. Finally, several subunits of ATP synthase were more highly expressed in the CKO mice. ATP synthase is referred to as the machine that makes ATP and hence plays a key role in biological energy metabolism [35–37]; specifically, it produces ATP from ADP using the proton gradient across the mitochondrial membrane that is generated by electron transport complexes of the respiratory chain [38]. Taken together, the above findings suggest a more active thermogenic gene program in the *Spop* CKO mice compared with the WT mice. Consistent with this conclusion, the cold assay revealed the *Spop* KO mice to display enhanced rates of oxygen consumption ($VO_2$) and $CO_2$ release ($VCO_2$), further implying a higher metabolic activity in the *Spop* CKO mice.

SPOP has been a frequently mutated hotspot, especially in many cancer types. Mutations that affect evolutionarily conserved residues in the substrate-binding MATH domain are predicted to result in the dysfunction of SPOP. The development of a specific SPOP inhibitor that would efficiently inhibit the ubiquitination and degradation of tumor repressors in the kidneys. Based on the current study, it would be reasonable to speculate that the "loss of function" mutations of and inhibition of SPOP might lead to reduced adipogenesis. However, further insights into the mechanistic basis of SPOP mediated regulation of adipogenesis and thermogenesis, especially in human and other animals, are still needed to explore the therapeutic potential of targeting SPOP.

## Materials and methods

### Ethics statement

Animal care and use protocols were performed in accordance with the regulations in the Guide for the Care and Use of Laboratory Animals issued by the Ministry of Science and Technology of the People's Republic of China. The animal experiments were approved by the Animal Ethics Committee of the Institute of Animal Sciences, Chinese Academy of Agricultural Sciences (Approval Number: IAS2018-8).

### Animals

*Spop* conditional knockout mice (*Spop*^F/F^) were created as previously described [19]. *Fstl1-Cre*^ERT2^ knock-in mice were obtained from the National Resource Center for Mutant Mice at Nanjing University. *Spop*^F/F^*Cre*^ERT2+/+^ mice were obtained by crossing of the above-mentioned mice, and the tamoxifen-induced knockout was operated as previously described [19]. Briefly, *Spop* conditional knockout mice were generated using Cre-LoxP recombination. Exons 4 and 5 of *Spop* were floxed with two *loxp* sites by CRISPR/Cas9-mediated targeting. Whole body expressed *Fstl1-Cre*^ERT2^ knock-in mice were obtained from the National Resource Center of China for Mutant Mice. *Spop*^F/F^*Cre*^ERT2+/+^ mice were obtained by crossing *Spop*^F/-^ and *Cre*^ERT2+/+^ mice. Mice were intraperitoneally injected with tamoxifen at the dose of 175 mg/kg bodyweight every other day three times to drive the expression of Cre. *Spop*^F/F^*Cre*^ERT2+/+^ mice without tamoxifen injection were used as the wild-type controls.

For the high-fat diet (HFD) treatment assay, an HFD (Research Diet, D12492) was fed to mice weighing at least 20 g for 15 weeks. In all other experiments, a chow diet was used to feed the mice.

### Cold treatment

Ten-week-old *Spop* knockout and control mice were subjected to a cold tolerance test in a 4˚C cold chamber for nine hours with free access to food and water. Mouse body temperature was measured using a rectal probe connected to a digital thermometer every hour.

### RNA-Seq analysis

Total RNA from BAT was isolated using TRIzol (Invitrogen) and then sequenced by an Illumina Novaseq platform. Low-quality reads were filtered, and clean reads were mapped to reference transcripts using Hisat2 V2.0.5. Differentially expressed genes were identified as having fold changes of $\leq -1.5$ or $\geq 1.5$, and $P < 0.05$ was considered to be significant.

### Proteomics analysis

Kidney tissue samples were ground with liquid nitrogen into powder, lysed in RIPA buffer, and then sonicated three times on ice using a high-intensity ultrasonic processor (Scientz). The supernatant was retained and the protein concentration was determined with a BCA kit according to the manufacturer's instructions. For mass spectrometry analysis, protein samples were firstly reduced and then digested by trypsin. Briefly, the tryptic peptides were dissolved in solvent A (0.1% formic acid, 2% acetonitrile/in water) and directly loaded onto a home-made reversed-phase analytical column. Peptides were then separated in solvent B. The peptides were subjected to capillary source followed by the timsTOF Pro (Bruker Daltonics) mass spectrometry.

### Western blot analysis

For western blot analyses, protein lysates or immunoprecipitate samples were separated by electrophoresis on SDS-PAGE gels and then transferred onto polyvinylidene fluoride membranes (Millipore). The membranes were first blocked with 5% (w/v) fat-free milk in TBST, then incubated with the corresponding primary antibodies diluted in 5% fat-free milk in TBST. After washing with TBST, the membranes were incubated with the appropriate secondary antibodies diluted in 5% fat-free milk in TBST. The protein bands were visualized using Immobilon Western Chemiluminescent HRP Substrate (Millipore) according to the manufacturer's instructions.

### Metabolic response

Oxygen consumption ($VO_2$) was measured using the Comprehensive Laboratory Animal Monitoring System (CLAMS) (Colombus Instruments). Data were normalized to lean body mass as determined by EchoMRI. Mice were individually caged and maintained under a 12-hour light/12-hour dark cycle.

### Citrate synthase activity measurement

Citrate synthase activity was measured using the MitoCheck Citrate Synthase Activity Assay Kit (Caymen Chemical) according to the manufacturer's instructions.

### Primary adiopocytes differentiation and oil red O staining

Inguinal white adipose tissue was excised from 8-week-old mice and digested with collagenase type I to prepare mouse preadiopocytes. Preadiopocytes were cultured in DMEM/F12 supplemented with 10% FBS and 1% penicillin/streptomycin. two days after reaching confluence, the cells were treated with culture medium supplemented with 5 μg/mL insulin, 1 μmol/L dexamethasone and 0.5 mmol/L 3-isobutyl- 1-methylxanthine to induce differentiation. Two days later, the cells were transferred to culture medium containing 5 μg/mL insulin and 1 μmol/L rosiglitazone for 2 days and then maintained in culture medium for 8 days.

For oil red O staining, the cells were firstly washed twice with PBS and fixed in 4% paraformaldehyde at room temperature. A 60% isopropanol solution was used to wash the cells after twice washing with distilled water. The cells were then stained with freshly prepared oil red O for 10 min and then washed with distilled water and photographed using a microscope.

### Statistical analysis

Statistical comparisons between two groups were made using the two-tailed Student's t test; comparisons between three or more groups were made by two-way ANOVA. All results are presented as the mean ± s.e.m. In all analyses, $p < 0.05$ was considered statistically significant (*, $p < 0.05$; **, $p < 0.01$; ***, $p < 0.001$).

## Supporting information

**S1 Fig. The feed intake amounts of the CKO and wild-type mice feed with a chow diet.**
(PDF)

**S2 Fig. KEGG pathway analysis of differentially expressed genes uncovered by RNA-seq between the CKO and wild-type controls.**
(PDF)

**S3 Fig. The feed intake amounts of the CKO and wild-type mice feed with an HFD diet.**
(PDF)

**S4 Fig. The relative mRNA levels of FASN and PGC1a in the BAT.**
(PDF)

**S5 Fig. The RER of the CKO and wild-type mice at the indicated times under normal conditions.**
(PDF)

**S1 Table. Differentially expressed proteins in the kidney of the *Spop* CKO mice using 4D label-free quantitative proteomics.**
(XLSX)

**S2 Table. Differentially expressed genes in the BAT of the *Spop* CKO mice identified by RNA-seq.**
(XLSX)

**S3 Table. Differentially expressed proteins in the liver of the *Spop* CKO mice feed with an HFD diet.**
(XLSX)

## Author Contributions

**Conceptualization:** Qinghe Li, Yuanyuan Wang, Guiping Zhao, Li Chen, Yu Gao.

**Data curation:** Qinghe Li, Yuhong Liu.

**Formal analysis:** Qinghe Li, Yu Gao.

**Funding acquisition:** Qinghe Li, Yuanyuan Wang, Li Chen, Yu Gao.

**Investigation:** Qinghe Li, Yuhong Liu, Yuanyuan Wang, Qi Zhang, Na Zhang, Danli Song, Fei Wang, Qianmei Gao, Yuxin Chen, Gaomeng Zhang, Jie Wen, Yu Gao.

**Methodology:** Qinghe Li, Yuhong Liu, Na Zhang, Qianmei Gao, Jie Wen.

**Project administration:** Qinghe Li, Jie Wen, Guiping Zhao.

**Resources:** Qinghe Li.

**Supervision:** Qinghe Li, Yu Gao.

**Validation:** Na Zhang.

**Writing – original draft:** Qinghe Li, Yu Gao.

**Writing – review & editing:** Qinghe Li, Yu Gao.

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
