## [Decision Letter · Decision Letter 0]

3 Sep 2024

Dear Dr Li,

Thank you very much for submitting your Research Article entitled 'Spop Deficiency Impairs Adipogenesis and Promotes Thermogenic Capacity in Mice' to PLOS Genetics.

The manuscript was fully evaluated at the editorial level and by independent peer reviewers. The reviewers appreciated the attention to an important problem, but raised some substantial concerns about the current manuscript. Based on the reviews, we will not be able to accept this version of the manuscript, but we would be willing to review a much-revised version. We cannot, of course, promise publication at that time.

If you decide to revise the manuscript for further consideration at PLOS Genetics, please aim to resubmit within the next 60 days, unless it will take extra time to address the concerns of the reviewers, in which case we would appreciate an expected resubmission date by email to plosgenetics@plos.org.

To resubmit, log into your Editorial Manager account and select the option 'Revise Submission' in the 'Submissions Needing Revision' folder.

We are sorry that we cannot be more positive about your manuscript at this stage. Please do not hesitate to contact us if you have any concerns or questions.

Yours sincerely,

Wei Guo, PhD

Guest Editor

PLOS Genetics

Monica Colaiácovo

Section Editor

PLOS Genetics

Dear Dr. Gao,

Thank you for submitting your manuscript, titled "Spop Deficiency Impairs Adipogenesis and Promotes Thermogenic Capacity in Mice" to PLOS Genetics. After careful consideration of the reviews provided by two expert reviewers, the editors believe that your study presents interesting findings regarding the role of SPOP in adipogenesis and thermogenesis. However, additional work that provides more biological and mechanistic insight would be necessary for publication in PLOS Genetics.

We invite you to address the reviewers' concerns and revise your manuscript accordingly. Once revised, your manuscript will undergo further review to assess its suitability for publication.

Thank you for considering PLOS Genetics as a venue for your work. We look forward to receiving your revised submission.

Reviewer's Responses to Questions

**Comments to the Authors:**

Reviewer #1: As the adaptor protein that determines substrate specificity of the Cul3-SPOP-Rbx1 E3 ligase complex, SPOP is involved in numerous biological processes. However, its physiological connections with adipogenesis and thermogenesis remain poorly understood. In the current study, we report that conditional knockout of Spop in mouse results in substantial changes in protein expression, including the upregulation of a critical factor associated with thermogenesis, UCP1. Loss of SPOP also led to defects in body weight gain. In addition, conditional knockout mice exhibited resistance to high-fat-diet-induced obesity. Proteomics analysis found that proteins upregulated in the knockout mice are primarily enriched for functions in glycolysis/gluconeogenesis, oxidative phosphorylation, and thermogenesis. Furthermore, Spop knockout mice were more resilient in cold tolerance assay compared with the wild-type controls. Finally, knockdown of SPOP efficiently impaired adipogenesis in NIH3T3-L1 cells and the expression of associated genes. Collectively, these findings demonstrate critical roles of SPOP in regulating adipogenesis and thermogenic capacity in mouse. There are a number of issues for this work.

1. The manuscript heavily focuses on well-known pathways and mechanisms related to thermogenesis and adipogenesis without providing significant novel insights.

2. Metabolic cage study should be included. The study primarily uses a conditional knockout model and in vitro cell lines. While these models are valuable, the lack of additional validation in other models or human tissues limits the generalizability of the findings.

3. The manuscript lacks detailed information on the statistical methods used for data analysis.

4. While the study identifies key proteins and pathways affected by SPOP deficiency, it lacks in-depth mechanistic exploration of how these changes lead to the observed phenotypes. Enzymes for lipid synthesis and metabolism should be examined.

5. The manuscript discusses the findings in the context of obesity and metabolic diseases but does not sufficiently address the translational potential of these findings.

6. Some figures lack clarity and detailed explanations in the text.

7. There are grammatical errors and awkward phrasings throughout the manuscript.

8. There is some redundancy in the introduction and discussion sections, where similar points are reiterated. Streamlining these sections to avoid repetition would make the manuscript more concise and focused.

Reviewer #2: This manuscript reports altered adipogenesis and thermogenesis in mice lacking SPOP in Fstl1-expressing cells. While the phenotype is interesting, some limitations in the experimental design were noticed. Also, details of some experiments that are required to interpret the data are missing.

1. More details about the Cre line used in this study are needed. What cell types express Fstl1-CreERT2?

2. The study is about adipogenesis and thermogenesis. Why did the authors choose the kidney for the proteomics analysis and KO validation western blot in Figure 1? Changes in the kidney do indicate what happens in other tissues.

3. What is the genotype of WT mice used in this study?

4. Supplemental figure legends are missing.

5. Figure 3 shows a higher relative weight for all tested organs in KO vs. WT. Which organ(s) in the KO have reduced weight compared to WT to cause less body weight gain in these mice? What are the organs’ absolute weights?

6. Quantifications for all western blots and histological images are needed.

7. Lines 182-184. “Wild-type mice were substantially more cold-tolerant compared with the CKO mice, as their body temperature started to drop significantly after 3 hr of cold challenge (Fig 4a).” Is this what the authors are trying to describe?

8. Figure 5. The efficiency of the SPOP knockdown needs to be validated. Also, the experiment is better done using primary cells from WT and KO mice.

**Have all data underlying the figures and results presented in the manuscript been provided?**

Reviewer #1: Yes

Reviewer #2: **No: **This study include RNAseq and proteomics data. However, the manuscript provides no access to the raw data.

PLOS authors have the option to publish the peer review history of their article (what does this mean?). If published, this will include your full peer review and any attached files.

Reviewer #1: **Yes: **JUN REN

Reviewer #2: No

---

## [Editor Report · Decision Letter 1]

26 Nov 2024

Dear Dr Li,

We are pleased to inform you that your manuscript entitled "Spop Deficiency Impairs Adipogenesis and Promotes Thermogenic Capacity in Mice" has been editorially accepted for publication in PLOS Genetics. Congratulations!

Before your submission can be formally accepted and sent to production you will need to complete our formatting changes, which you will receive in a follow up email. Please be aware that your manuscript is conditionally accepted since the dataset is not yet publicly available and the accession number is not added to the manuscript. Therefore, the acceptance is conditional upon the successful submission of the dataset to a public repository and inclusion of the accession number in the manuscript. Please note: the accept date on your published article will reflect the date of this provisional acceptance, but your manuscript will not be scheduled for publication until the required changes have been made.

Yours sincerely,

Wei Guo, PhD

Guest Editor

PLOS Genetics

Monica Colaiácovo

Section Editor

PLOS Genetics

Aimée Dudley

Editor-in-Chief

PLOS Genetics

Anne Goriely

Editor-in-Chief

PLOS Genetics

Comments from the reviewers (if applicable):

**Data Deposition**

http://datadryad.org/submit?journalID=pgenetics&manu=PGENETICS-D-24-00780R1

**Press Queries**

---

## [Editor Report · Acceptance letter]

3 Dec 2024

PGENETICS-D-24-00780R1 

Spop Deficiency Impairs Adipogenesis and Promotes Thermogenic Capacity in Mice 

Dear Dr Li, 

We are pleased to inform you that your manuscript entitled "Spop Deficiency Impairs Adipogenesis and Promotes Thermogenic Capacity in Mice" has been formally accepted for publication in PLOS Genetics! Your manuscript is now with our production department and you will be notified of the publication date in due course.

With kind regards,

Anita Estes

PLOS Genetics

On behalf of:
